# MicroRNA-631 Resensitizes Doxorubicin-Resistant Chondrosarcoma Cells by Targeting Apelin

**DOI:** 10.3390/ijms24010839

**Published:** 2023-01-03

**Authors:** Jui-Chieh Chen, Hsun-Chang Shih, Chih-Yang Lin, Jeng-Hung Guo, Cheng Huang, Hsiu-Chen Huang, Zhi-Yong Chong, Chih-Hsin Tang

**Affiliations:** 1Department of Biochemical Science and Technology, National Chiayi University, Chiayi 600355, Taiwan; 2Department of Anesthesiology, Ditmanson Medical Foundation Chia-Yi Christian Hospital, Chiayi 60002, Taiwan; 3Translational Medicine Center, Shin-Kong Wu Ho-Su Memorial Hospital, Taipei 111045, Taiwan; 4Graduate Institute of Biomedical Science, China Medical University, Taichung 404333, Taiwan; 5Department of Neurosurgery, China Medical University Hospital, Taichung 404333, Taiwan; 6Department of Biotechnology and Laboratory Science in Medicine, National Yang Ming Chiao Tung University, No. 155, Sec. 2, Linong St., Beitou District, Taipei 11221, Taiwan; 7Center for Teacher Education, National Tsing Hua University, Hsinchu 300, Taiwan; 8Department of Applied Science, Nanda Campus, National Tsing Hua University, Hsinchu 300, Taiwan; 9Department of Pharmacology, School of Medicine, China Medical University, Taichung 404333, Taiwan; 10Chinese Medicine Research Center, China Medical University, Taichung 404333, Taiwan; 11Department of Biotechnology, College of Health Science, Asia University, Taichung 400354, Taiwan

**Keywords:** chondrosarcoma, doxorubicin, apelin, microRNA, drug resistance

## Abstract

Chondrosarcoma is the second most common type of bone cancer. Surgical resection is the best choice for clinical treatment. High-grade chondrosarcoma is destructive and is more possible to metastasis, which is difficult to remove using surgery. Doxorubicin (Dox) is the most commonly used chemotherapy drug in the clinical setting; however, drug resistance is a major obstacle to effective treatment. In the present study, we compared Dox-resistant SW1353 cells to their parental cells using RNA sequencing (RNA-Seq). We found that the apelin (APLN) pathway was highly activated in resistant cells. In addition, tissue array analysis also showed that APLN was higher in high-grade tissues compared to low-grade tissues. APLN is a member of the adipokine family, which is a novel secreted peptide with multifunctional and biological activities. Previously, studies have shown that inhibition of the APLN axis may have a therapeutic benefit in cancers. However, the role of APLN in chondrosarcoma is completely unclear, and no related studies have been reported. During in vitro experiments, APLN was also observed to be highly expressed and secreted in Dox-resistant cells. Once APLN was knocked down, it could effectively improve its sensitivity to Dox. We also explored possible upstream regulatory microRNAs (miRNAs) of APLN through bioinformatics tools and the results disclosed that miR-631 was the most likely regulator of APLN. Furthermore, the expression of miR-631 was lower in the resistant cells, but overexpression of miR-631 in the Dox-resistant cell lines significantly increased the Dox sensitivity. These results were also observed in another chondrosarcoma cell line, JJ012 cells. Taken together, these findings will provide rationale for the development of drug resistance biomarkers and therapeutic strategies for APLN pathway inhibitors to improve the survival of patients with chondrosarcoma.

## 1. Introduction

Chondrosarcoma, a malignant cartilage neoplasm, is the second most common bone cancer. It is resistant to conventional chemotherapy and radiotherapy, and surgical resection is still the main clinical treatment [1]. The histological classification can be divided into three grades (grades I-III), with different levels of cellular morphology, nuclear heterogeneity, myxoid, and angiogenesis [2,3]. Grade III chondrosarcoma occurs in barely approximately 5–10% of patients, but it has a high metastatic potential, which is difficult to remove surgically, and ultimately leads to patient death. Thus, the development of new therapeutic strategies is urgently needed [4,5].

Doxorubicin (Dox) with a four-membered ring is an anthracycline anticancer antibiotic drug [6] that has been widely used in the treatment of various cancers [7,8]. In soft tissue sarcoma, Dox is one of the most effective chemotherapeutic drugs [9,10,11]. The clinical treatment of chondrosarcoma with Dox can significantly improve the survival rate of patients [12]. However, some patients may gradually develop resistance to the drug after a period of treatment, but if the dose of treatment is increased, it will cause cardiac toxicity [13]. The development of Dox combined with multiple drugs or molecules for co-treatment can reduce its toxicity and improve the response rate of treatment [14].

Apelin (APLN), an evolutionarily conserved secreted protein, is a member of the adipokine family. APLN is first synthesized as a 77-amino acid prepropeptide, which is cleaved to form the mature secreted form of 36-amino acid peptide. It is then engaged to its G protein-coupled apelin receptor (APLNR) to activate downstream pathways [15]. APLN and APLNR can be expressed in various organs and tissues in the human body, and have been found to be expressed in the cardiovascular system, central nervous system, digestive system, circulatory system, reproductive system, muscular system, and adipose tissue [16]. During normal blood vessel development, APLN can act as an active substance to stimulate the proliferation and migration of endothelial cells [17,18]; therefore, APLN has also been confirmed as a potentially important factor in promoting tumor angiogenesis [19]. In addition, APLN and its receptors are involved in signaling pathways related to migration and invasion, leading to tumor growth and metastasis [20]. Studies have shown that inhibition of the APLN–APLNR axis by a pharmacological targeting against APLN can reduce tumor growth [21,22]. Accumulating studies have shown that APLN may be a promising biomarker to predict therapy response and survival prognosis in patients [23,24,25]. 

MicroRNAs (miRNAs) are endogenous noncoding RNAs of approximately 22 nucleotides that can regulate physiological and pathological processes by binding to complementary sequences in mRNA targets to inhibit their protein expression [26]. It is now well known that these tiny miRNAs act as important gene regulators, in which up to a third of the human genes are modulated by miRNAs [27]. MiRNAs are, therefore, key regulators of numerous genes to control various biological processes such as cell growth, differentiation, and death [28]. In cancer, miRNAs are also shown to regulate an array of cellular processes such as cell proliferation, cell cycle, differentiation, migration, angiogenesis, apoptosis, and drug resistance leading to cancer progression [29,30]. 

In the present study, we established Dox-resistant cells to analyze the differential gene expression using RNA-Seq and found that the APLN pathway appears to be involved in Dox resistance. However, the functions and related mechanisms of APLN and APLN-related miRNAs have not been reported in chondrosarcoma. This study elucidated the relationship between the miRNA–APLN axis and Dox resistance in chondrosarcoma cells. These results may aid in the development of APLN pathway inhibitors in combination with Dox to resensitize resistant cells and serve as the basis for the development of resistance biomarkers.

## 2. Results

### 2.1. Activation of APLN Pathway May Be Associated with Dox Resistance

To explore the possible mechanism of Dox resistance in chondrosarcoma cells, a Dox-resistant SW1353 subline designated SW1353R was established by gradually increasing the drug concentration from 0.0625 μM to 2.0 μM for approximately six months. As shown in Figure 1A, the results of flow cytometry analysis showed that SW1353R cells were only approximately 10% cell death after treatment with 2 μM Dox for 24 h, while the parental SW1353 cells were more than 80% cell death, indicating that SW1353R is highly resistant to Dox. To identify candidate genes involved in Dox resistance in chondrosarcoma, we isolated RNA from SW1353 and SW1353R cells for RNA-Seq transcriptome analysis. A total of 1438 upregulated genes in SW1353R cells were identified using a threshold of |log_2_fold change (FC)| > 2 and padj value < 0.05. These genes were then subjected to bioinformatics analysis using the ShinyGO web server. The GO enrichment was obtained for the exploration of biological process, cellular component, molecular function, and KEGG using a false discovery rate (FDR) < 0.05. The most significant biological processes for upregulated genes involved neurogenesis, development, and differentiation; cell-cell signaling; and cell migration, organogenesis, and development (Figure 1B). The most significant cellular components for upregulated genes involved membrane protein components; nerve-related components; and extracellular matrix components (Figure 1C). The most significant molecular functions for upregulated genes involved signal receptor binding; DNA binding and regulation of gene expression; calcium ion binding; ion channel activity; extracellular matrix function; glycosaminoglycan binding; and phosphodiester hydrolase activity (Figure 1D). The most significant KEGG for upregulated genes involved in pathways in cancer were the PI3K-Akt signaling pathway; axon guidance; focal adhesion; calcium signaling pathway; RAS signaling pathway; cAMP signaling pathway; Wnt signaling pathway; cGMP-PKG signaling pathway; protein digestion and absorption; oxytocin signaling pathway; hippo signaling pathway; insulin secretion; glutamatergic synapse; APLN signaling pathway; ECM–receptor interaction; TGF-beta signaling pathway; circadian entrainment; gastric acid secretion; and ABC transporters (Figure 1E). In addition, a KEGG network was also constructed, and the results showed that the APLN signaling pathway has the highest relevant connectivity (Figure 1F).

### 2.2. APLN Was Highly Expressed and Secreted in SW1353R Cells, and Knockdown of APLN Can Increase Dox Sensitivity in SW1353R Cells; APLN Was Also Associated with Clinical Progression in Chondrosarcoma

To further explore the relationship between APLN and Dox resistance, we performed qPCR and Western blot to determine if APLN was highly expressed in SW1353R cells. As shown in Figure 2A,B, the results indicated that the expression of APLN was much higher in SW1353R cells than SW1353 cells at mRNA and protein levels. In addition, we also used Western blot to analyze the amount of APLN secretion in conditioned media between SW1353 and SW1353R cells. As shown in Figure 2C, SW1353R cells had higher APLN secretion than SW1353 cells. These data indicated that high expression and secretion of APLN in chondrosarcoma cells may be involved in Dox resistance. To determine the effects of APLN expression on Dox sensitivity, we suppressed APLN expression by lentivirus-mediated RNAi system. The knockdown efficiency of APLN was evaluated by Western blot (Figure 2D). Knockdown APLN expression resulted in elevated sensitivity to Dox in a dose-dependent fashion, as assessed by MTT assay (Figure 2E). Moreover, knockdown of APLN in SW1353R cells significantly promoted Dox-induced apoptosis, as evidenced by the increase in the cleavage of caspase 3 (Figure 2F). To determine the correlation between APLN expression in chondrosarcoma clinical tissues and its progression, we analyzed APLN expression in low-grade and high-grade groups using tissue arrays. The results revealed that the expression level of APLN was higher in the high-grade tissues compared with low-grade tissues (Figure 2G).

### 2.3. miR-631 Can Target APLN to Reduce Its Protein Expression to Restore Dox Sensitivity in SW1353R Cells

MiRNAs are important regulators of distinct biological processes and are tightly related to a variety of diseases including drug resistance in cancer. Therefore, we wondered whether the expression of APLN is mediated by miRNAs to alter the Dox sensitivity in chondrosarcoma. To search for putative miRNAs that could target the APLN mRNA, we first combined putative miRNAs targeting APLN from different databases. Computational algorithms miRTarBase 7.0, mirsearch V3.0, miRDB, miR system, TarBase v.8, miRmap, and mirwalk were used to identify miRNAs that could potentially regulate APLN expression. The result showed that five of the seven databases predicted that miR-631 can target APLN (Figure 3A). As shown in Figure 3B, the 3’ UTR of APLN has multiple regions that can be bound and regulated by miR-631. Next, we analyzed the expression of miR-631 in SW1353 and SW1353R cells using qPCR, and the results showed that SW1353R had significantly lower miR-631 expression than its parental cells (Figure 3C). Therefore, we hypothesized that upregulation of miR-631 in SW1353R cells may lead to decreased protein levels of APLN and restore its sensitivity to Dox. SW1353R cells were transfected with miR-631 mimic and its control (miR-NC). The results of qPCR analysis demonstrated that SW1353R cells significantly overexpressed miR-631 compared with the miR-NC group (Figure 3D). Furthermore, we also analyzed the RNA and protein level of APLN using qPCR and Western blot and the results showed that overexpression of miR-631 could effectively reduce the expression of APLN in SW1353R cells (Figure 3E,F). To determine whether miR-631 overexpression could restore Dox sensitivity in SW1353R cells, we further dissected by the levels of cleaved caspase 3 and functional assays (MTT assay and LDH release). The results revealed that Dox sensitivity was significantly improved once SW1353R cells overexpressed miR-631 (Figure 3G–I).

### 2.4. High Expression of APLN Also Enhanced the Dox Resistance in JJ012 Cells, Whereas Overexpression of miR-631 Can Restore Dox Sensitivity

In addition, we investigated whether APLN has a similar effect on Dox resistance in another chondrosarcoma cell line (JJ012 cells). First, we established Dox-resistant JJ012 (JJ012R) cells to study the role of APLN in Dox resistance. As shown in Figure 4A, JJ012R cells were more resistant to Dox than their parental JJ012 cells in a dose-dependent manner, as determined by Annexin V/PI staining assay. To further confirm the relationship between APLN and Dox resistance, we performed Western blot to determine if APLN is highly expressed in JJ012R cells. As shown in Figure 4B, the results indicated that expression of APLN was much higher in JJ012R cells than JJ012 cells. To further validate the effect of APLN expression on Dox sensitivity, APLN was knocked down in JJ012R cells. Western blot analysis observed that marked knockdown of APLN (Figure 4C) led to an increased sensitivity to Dox in a concentration-dependent manner, as analyzed by the MTT assay (Figure 4D). Moreover, APLN knockdown in JJ012R cells also significantly promoted Dox-induced apoptosis, as evidenced by the increase in the cleavage of caspase 3 (Figure 4E). Similarly, JJ012R cells were transfected with miR-631 mimic and analyzed whether it could restore Dox sensitivity. The results of qPCR analysis showed that JJ012R cells transfected with miR-631 mimic could highly express miR-631 compared with miR-NC (Figure 4F). In the case of miR-631overexpression, the RNA and protein expression of APLN was also dramatically decreased in JJ012R cells (Figure 4G,H), which rendered Dox-resistant cells more sensitive to Dox, as shown by MTT assay and LDH release assay (Figure 4I,J).

## 3. Discussion

In the treatment of cancer, the generation of drug resistance has always been a difficult problem. Thus, this is a major hurdle to successfully cure chondrosarcoma. In the present study, we found that APLN was highly expressed and secreted in Dox-resistant chondrosarcoma cells. Furthermore, knockdown of APLN can sensitize chondrosarcoma cells to Dox. Dox-resistant cells must undergo multiple mutations to adapt to Dox-containing environments, and APLN may be one of the pathways involved. Therefore, even APLN knockdown could not fully restore the same Dox sensitivity as their parental cells. Previously, studies indicated that the APLN–APLNR signaling nexus may become a signal to maintain the expansion and development of tumor cells. Pharmacologic targeting of the APLN–APLNR axis can reduce the proliferation and growth of tumor cells, which may be of therapeutic benefit in cancer [21,22,31]. Therefore, APLN pathway inhibitors may also be applied in the treatment of chondrosarcoma in the future to improve the survival rate of patients.

Accumulating studies have shown that APLN may be a promising biomarker to predict therapy response and survival prognosis in patients. In colorectal cancer, high APLN protein expression was correlated with poor progression-free survival in bevacizumab-treated patients [32]. In gastric cancer, the overall survival of patients with high expression of APLN in tumor was shorter than those with low expression [33]. In non-small-cell lung carcinoma (NSCLC), APLN mRNA levels were significantly increased in tumor samples compared with normal lung tissue, and high APLN protein levels were associated with elevated microvessel densities and poor overall survival [34]. In bladder cancer, APLN was upregulated in tumor tissues compared with matched adjacent noncancer tissues, especially in the high tumor stage, distant metastasis, and vascular invasion [35]. In prostate cancer, upregulation of APLN is more frequently occurred in tumor tissues with advanced pathologic stage, metastasis, prostate-specific antigen failure, and shorter biochemical recurrence-free survival [25]. Thus, inhibition of the APLN–APLNR axis may become a new therapeutic target for a variety of cancers, including chondrosarcoma.

APLN, a novel, multifunctional, and bioactive secreted peptide, is an evolutionarily conserved adipokine. It can act as the endogenous ligand for the orphan G protein-coupled receptor called APLNR, which was originally named APJ [15]. APLN is derived from a 77-amino acid precursor that could be cleaved into apelin-36 or shorter bioactive peptides such as apelin-13 and -17 [36]. It was observed that this bioactive peptide stimulates proliferation and migration of endothelial cells and is required for normal vascular development [17,18]. In addition, APLN and its receptors are involved in signaling pathways related to migration and invasion, leading to tumor growth and metastasis [20].

It has been reported that the APLN–APLNR signaling nexus promotes cancer development through several mechanisms, that is, cell proliferation, development of cancer stem cells, drug resistance, and inhibition of cancer cell apoptosis [37]. Pharmacologic targeting of the APLN–APLNR axis can reduce the proliferation and growth of tumor cells, which may be of therapeutic benefit in cancers [21,22,31]. Many studies reported that apelin-13 can stimulate tumor proliferation and metastasis [38]. In addition, many studies have shown that the regulation of the APLN pathway can be a therapeutic target, and APLNR antagonists can be used to treat a variety of diseases [39]. APLN has been recognized for its emerging role as a cancer treatment target [40]. APLR antagonists, including ML 221, MM 54, and protamine, have been shown to effectively block the activation of the APLN axis and thus affect cell functions. The U.S Food and Drug Administration-approved compound, protamine, is usually used after cardiac surgery, where it binds to heparin to reverse anticlotting activity. Most recently, protamine has been identified as an APLNR antagonist, which can explain some of the side effects observed in patients treated with protamine, but this means that anti-angiogenic activity may be used to treat cancer [41].

The complex microenvironment of tumors involves interactions between different types of cells. Secreted proteins are responsible for crosstalk between cells, which may contribute to tumor resistance [42,43,44,45]. In addition, secreted proteins are excellent candidates for serological tumor markers, since secreted proteins are released by cells and have a very high probability of entering the circulation [46,47]. Since APLN is a secreted protein, it may be secreted into the serum, and it has the opportunity to be used as a clinical biomarker in the future. Similar results have been reported in serum of patients with a significant increase in APLN levels, which is also related to the poor prognosis of cancer. In gastroesophageal cancer, patients’ APLN levels were significantly higher in tumor tissue and serum than in healthy controls [48]. In esophageal squamous cell carcinoma (ESCC), APLN levels were significantly higher in tumor tissue and serum than in healthy controls [49]. In head and neck cancer (HNC), serum APLN levels were significantly higher in patients than in the healthy controls; however, APLN was significant decreased after radiotherapy [50]. To date, biomarkers for drug resistance in chondrosarcoma remain ill-defined.

However, it is largely unclear whether miRNAs can increase Dox sensitivity by regulating APLN in chondrosarcoma. A previous study has shown that targeting APLN by miRNA can inhibit the invasion and migration of prostate cancer cells, which may synergistically predict biochemical recurrence-free survival in patients [25]. More recently, another study indicated that miRNA inhibits lung adenocarcinoma cell proliferation and invasion though targeting APLN and provides novel insight into the mechanism underlying the development of lung adenocarcinoma [51]. Furthermore, many of these adipokine-regulated miRs have been characterized for their relevance on tumor biological functions, including proliferation, invasion, apoptosis inducibility, and angiogenesis. Subsequently, some of these adipokine-regulated miRs can be characterized as oncogenic or anti-tumoral miRs [21,52]. Since miRNAs can act as oncogenes or tumor suppressor genes, they play vital biological roles in carcinogenesis [53]. Recently, some scientists proposed the concept of “Pharmaco-miR” from the perspective of miRNA drugs [54]. In addition to predicting drug behavior, miRNA can also improve the efficacy of drugs.

## 4. Materials and Methods

### 4.1. Cell Culture and Establishment of Dox-Resistant Cells

The human chondrosarcoma cell line SW1353 was obtained from the American Type Culture Collection (ATCC), and JJ012 was kindly provided by the laboratory of Dr. Sean P. Scully (University of Miami School of Medicine, Miami, FL). Cells were cultured in Dulbecco’s modified Eagle’s medium (DMEM) containing 10% FBS and 1% penicillin/streptomycin, and then plated in an incubator with humidified conditions at 37 °C with 5% CO_2_. Dox-resistant cell lines SW1353R and JJ012R were treated with stepwise increasing Dox concentrations to 2 µM and 8 µM, respectively, until cells had a stable resistant phenotype. For analysis of secreted proteins, cells were grown in serum-free DMEM, and conditioned medium was collected after 48 h. Culture supernatants were then concentrated with 3 kDa-cutoff Amicon Ultra (Merck KGaA, Darmstadt, Germany).

### 4.2. Apoptosis Evaluated by Flow Cytometry

Cells (3 × 10^5^ cells/well) were seeded into 6-well plates and incubated overnight to allow them to adhere. Cells were stimulated with the indicated doses of Dox for 24 h and stained with Annexin-V-FITC and PI (BD Biosciences, San Jose, CA, USA). The percentage of apoptosis was then determined using flow cytometry. The upper right quadrant (Annexin V+/PI+) was defined as late apoptotic cells for analysis.

### 4.3. RNA Sequencing and Enrichment Analyses of GO and KEGG 

Total RNA was extracted using REzol reagent (Protech Technology Enterprise CO., Ltd., Taipei, Taiwan). RNA library construction and sequencing were conducted by the Genomics Co., Ltd. (New Taipei City, Taiwan). The NovaSeq 6000 platform (Illumina, San Diego, CA, USA) was applied to produce 150 bp paired-end reads. TPM (Transcripts Per Million) of each gene was calculated using RSEM. Differential expression analysis was determined using DESeq2 package. We utilized a |log_2_fold change (FC)| > 2 with adjusted *p* (*p*adj) value < 0.05 as the threshold for significantly differential expression. Subsequently, we used ShinyGO to analyze differentially expressed genes in SW1353R cells higher than those in SW1353 cells, their biological processes, cellular components, molecular functions, and KEGG pathways. The KEGG network was used to present potential protein–protein interactions between components of the KEGG pathway.

### 4.4. Tissue Arrays

After tissue sections were deparaffinized in xylene, endogenous peroxidase was inhibited with 5% hydrogen peroxide for 20 min. The slide was incubated with 3% BSA (dissolved in tris-buffered saline containing 0.1% triton X-100) for 1 h. The sections were then incubated with primary antibodies against APLN (1:250 dilution; for tissue array cat. no. OASG00449, Aviva system biology) at 4 °C for overnight. Unbound antibody was washed away, and then slides were probed with biotinylated anti-rabbit secondary antibody for 1 h, and then reacted with peroxidase-conjugated streptavidin for 1 h. DAB substrate was added to generate the brown chromogen a−/n+ d hematoxylin which was used for counterstaining.

### 4.5. qPCR

Total RNAs were isolated using REzol reagent. The mRNA was reverse transcribed to cDNA using the PrimeScript RT Reagent kit (Takara Bio, Inc., Shiga, Japan), and subsequently the KAPA SYBR FAST qPCR Master Mix (Takara Bio, Inc., Shiga, Japan) was used to detect APLN levels. The Mir-X miRNA qRT-PCR SYBR kit (Clontech Laboratories, Inc., CA, USA) was used to detect miR-631 levels. The MyGo PCR Detection System (IT-IS Life Science Ltd., Dublin, Ireland) was used for detection. The primers used for the experiment were as follows: APLN (forward primer: 5′-GTCTCCTCCATAGATTGGTCTGC-3′; reverse primer: 5′-GGAATCATCCAAACTACAGCCAG-3′) and GAPDH (forward primer: 5′-CACCCATGGCAAATTCCATGGCA-3′; reverse primer: 5′-TCTAGACGGCAGGTCAGGTCCACC-3′), miR-631 (forward primer: 5′-AGACCTGGCCCAGACCTCAGC-3′); reverse primer: mRQ 3′ primers supplied with the kit. Relative gene expression was calculated using the comparative Ct (2^−ΔΔCT^) method with genes normalized to GAPDH (mRNA) or U6 (microRNAs).

### 4.6. Western Blot

Cells were lysed with RIPA buffer containing protease inhibitor and phosphatase inhibitor cocktail. The concentrations of protein were determined by the Pierce BCA Protein Assay Kit (Thermo Fisher Scientific Inc., Waltham, MA, USA). Equal concentrations of total protein were separated by SDS-PAGE and transferred to PVDF membranes (Millipore, Bedford, MA, USA). The membrane was blocked with 5% non-fat milk (dissolved in Tris-buffered saline) for 1 h. The primary antibodies were added and incubated at 4 °C for overnight. After being washed in Tris-buffered saline with 0.1% Tween-20 (TBST), the membranes were probed with appropriate HRP-conjugated secondary antibodies at RT for 1 h. The bound antibodies were detected using ECL reagents (Merck-Millipore) and autoradiography. The following antibodies were used: anti-tubulin (1:10,000; cat. no. 05–829; EMD Millipore), cleaved caspase-3 (1:1000; cat. no. #9661; Cell signaling), APLN (1:1000; cat. no. OASG00449; Aviva system biology), HRP-conjugated goat anti-rabbit IgG (1:5000; cat. no. 20202; Leadgene Biomedical, Inc., Tainan, Taiwan), and goat anti-mouse IgG (1:5000; cat. no. 115-035-003; Jackson ImmunoResearch Laboratories, Inc., West Grove, PA, USA).

### 4.7. Gene Knockdown

The pLKO.1-puro-based lentiviral vectors: target shAPLN#1 RNA (TRCN0000358700), shAPLN#2 RNA (TRCN0000358701), and pLKO.1-shScramble (TRC2) were obtained from National RNAi Core Facility at Academia Sinica, Taipei, Taiwan. Recombinant lentiviruses were packaged according to the manufacturer’s instruction. Cultured cells were incubated with lentiviral supernatants containing 8 μg/mL polybrene for 24 h, replaced with fresh medium, and incubated for another 48 h. For stable cell lines, cells were selected by puromycin (5 μg/mL).

### 4.8. MTT Assay

The 96-well flat bottom plates were seeded with 5 × 10^3^ cells in 100 μL of medium, incubated overnight to allow attachment, replaced with serum-free DMEM, and treated with different concentrations of Dox for 24 h. Before the end of the treatment period, MTT reagent (0.5 mg/mL) was added to each well and incubated for 4 h. The supernatant was aspirated and DMSO was added to dissolve the formazan crystals. The absorbance of the signals was measured at a wavelength of 550 nm and the background value was subtracted at 750 nm.

### 4.9. Bioinformatics Analysis

Prediction of miRNA binding to the APLN was performed by seven computer-aided algorithms: miRTarBase 7.0, mirsearch V3.0, miRDB, miR system, TarBase v.8, miRmap, and mirwalk.

### 4.10. Cell Transfection with miRNA Mimic 

Cells (2 × 10^5^ cells/well) were seeded into 6-well plates, and then hsa-miR-152-3p miRNA mimic and miRNA mimic negative controls (NC) (mirVana, Thermo Fisher Scientific) were transfected into cells with TurboFect transfection reagent at a final concentration of 25 nM.

### 4.11. Lactate Dehydrogenase (LDH) Assay

Cell death was also assessed by the LDH cytotoxicity assay kit (Sigma-Aldrich). Cells were stimulated with Dox for 24 h, and the culture supernatants were transferred to a 96-well plate. In addition, the total cells were lysed as maximum absorbance and culture medium was the control absorbance. Subsequently, the reaction mixture was added into a 96-well plate and incubated for 30 min at room temperature. After adding the stop solution, OD values was measured at 500 nm and 700 nm using a spectrophotometer. Cell death ratio was calculated using the following formula: Cell death (%) = (Absorbance sample − Absorbance control)/(Absorbance maximum − Absorbance control) × 100.

## 5. Conclusions

In the present study, we explored the mechanisms of Dox resistance in chondrosarcoma cells by RNA-Seq. The results indicated that activation of the APLN pathway may contribute to Dox resistance in chondrosarcoma cells. The expression of APLN was also observed to be related to the malignancy of clinical chondrosarcoma tissue. However, the role of APLN and its related regulation in chondrosarcoma is completely unknown, and no related studies have been reported. Therefore, we performed in vitro experiments to verify the role of APLN in Dox resistance of chondrosarcoma cells, and the results showed that APLN was highly expressed and secreted in Dox-resistant cells, and APLN knockdown could restore Dox sensitivity in Dox-resistant cells. Currently, miRNAs are emerging as a promising new application for cancer molecular diagnostics and therapeutic strategies. We also identified miR-631 as an upstream regulator of APLN. Overexpression of miR-631 suppressed APLN protein levels and further resensitized Dox-resistant cells to Dox. These results will help to develop a new biomarker for Dox resistance and to discover the new treatment remedies by blocking the activation of APLN in chondrosarcoma patients.

## Figures and Tables

**Figure 1 ijms-24-00839-f001:**
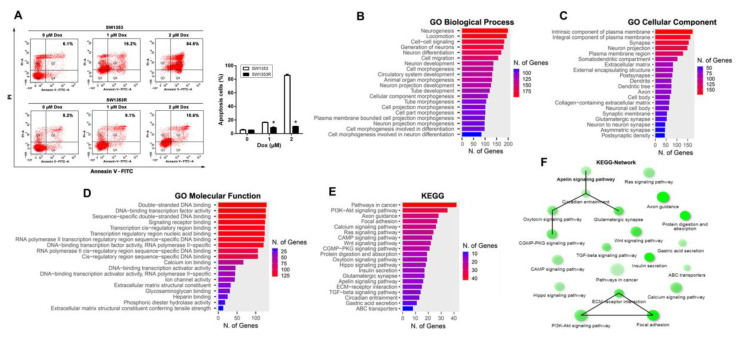
APLN was involved in chondrosarcoma progression. (**A**) SW1353 and SW1353R cells were treated with increasing doses of Dox (0–2 μM) for 24 h and cell were stained with annexin V-FITC and PI. Percentages of apoptosis cells were analyzed by flow cytometry. Data were expressed as mean ± SEM (*n* = 3). * *p* < 0.05 compared with SW1353 cells. (**B**–**E**) SW1353 and SW1353R cells were analyzed using RNA sequencing, and a total of 1438 genes were selected using thresholds of |log_2_fold change (FC)| > 2 and padj value < 0.05 for bioinformatics analysis of ShinyGO. Those genes were significantly upregulated in SW1353R cells compared to SW1353 cells. Bar plot shows the numbers of genes corresponding to their functional pathways by GO enrichment analysis. (**B**) Biological process of GO analysis; (**C**) cellular component of GO analysis; (**D**) molecular function of GO analysis; (**E**) KEGG pathways analysis. (**F**) Visualization of the relationship among KEGG pathways using network.

**Figure 2 ijms-24-00839-f002:**
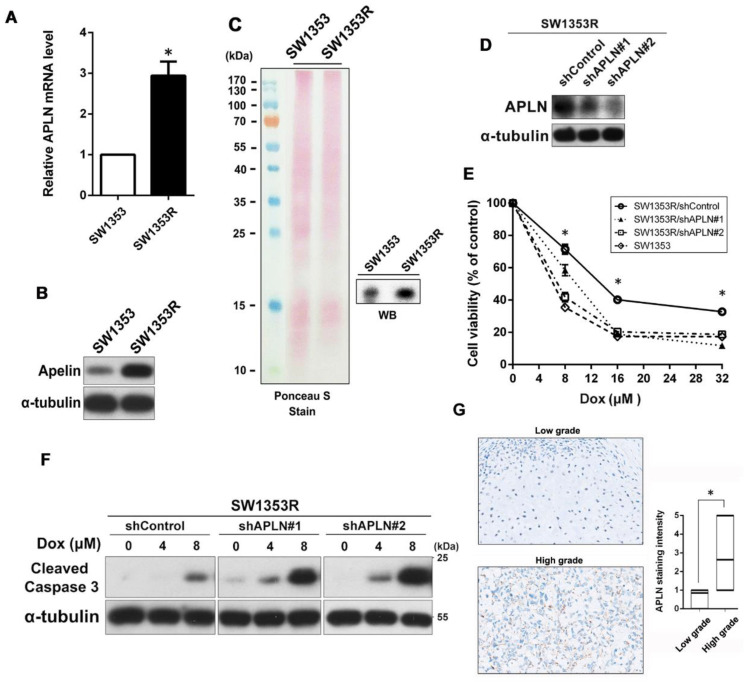
APLN was highly expressed and secreted in Dox-resistant SW1353R cells and reducing its expression enhanced sensitivity to Dox. (**A**) The expression levels of APLN mRNA were measured using qPCR. * *p* < 0.05 compared with SW1353 group. (**B**) The expression levels of APLN protein in total cell lysates were analyzed using Western blot. (**C**) The secretion of APLN in conditioned media (CM) from SW1353 and SW1353R cells was analyzed using Western blot. The amount of loading protein was evaluated using Ponceau S staining. (**D**) The protein levels of APLN in SW1353R/shControl, SW1353R/shAPLN#1, and SW1353R/shAPLN#2 cells were examined using Western blot. (**E**) Cells were exposed to increasing concentrations of Dox (0–32 μM) for 24 h, and subsequently cytotoxicity was evaluated by an MTT-based assay. The percentage of cell viability is shown relative to untreated controls. * *p* < 0.05 compared with shControl group. (**F**) Cells were treated with increasing concentrations of Dox (0–8 μM) for 24 h and the expression levels of cleaved caspase 3 were examined using Western blot. α-Tubulin was used as a loading control. (**G**) IHC staining for APLN in tissue microarray from low-grade chondrosarcoma (grade II, *n* = 13) and high-grade chondrosarcoma (grade III, *n* = 39). Scale bar = 100 μm.

**Figure 3 ijms-24-00839-f003:**
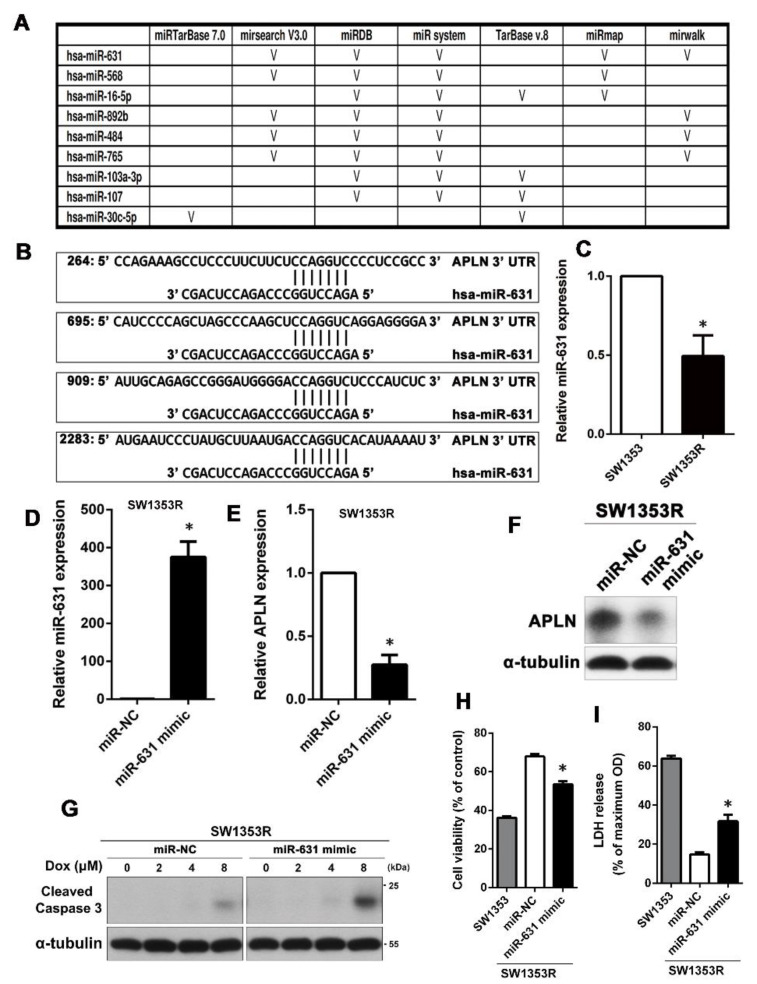
High expression of miR-631 can downregulate APLN protein levels to increase Dox sensitivity in SW1353R cells. (**A**) Seven different databases were combined to find microRNAs targeting APLN. (**B**) A possible miR-631 target region was found in the 3’ UTR of APLN mRNA. (**C**) The expression of miR-631 in SW1353 and SW1353R cells was analyzed using qPCR. * *p* < 0.05 compared with SW1353 cells. (**D**,**E**) SW1353R cells were transfected with miR-631 mimic or miR-NC and the expression levels of miR-631 and APLN were determined by qPCR. * *p* < 0.05 compared with miR-NC group. (**F**) After SW1353R cells were transfected with miR-631 mimic, the protein expression of APLN was analyzed by Western blot. (**G**) Cells were then stimulated with different concentrations of Dox (0–8 μM) for 24 h, and the cell viability was determined by MTT. (**H**,**I**) The effect of miR-631 on Dox sensitivity was also analyzed by MTT and LDH analyses. MiR-631-transfected SW1353R cells and their parental cells (SW1353 cells) were treated with 8 μM Dox for 24 h. * *p* < 0.05 compared with miR-NC group.

**Figure 4 ijms-24-00839-f004:**
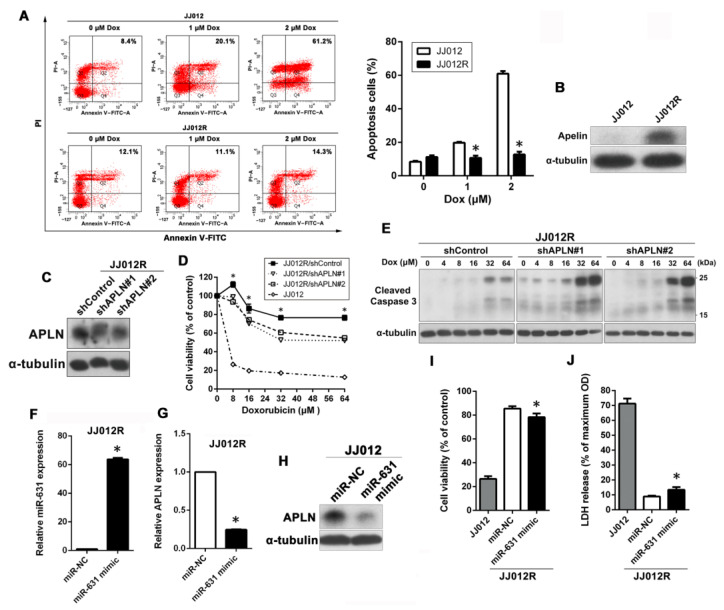
miR-631–APLN axis was also involved in Dox sensitivity in JJ012 cells. (**A**) Cells were exposed to increasing doses of Dox (0–2 μM) for 24 h and apoptotic cells were analyzed by the annexin V-FITC/PI approach. * *p* < 0.05 compared with JJ012 cells. (**B**) The expression levels of APLN protein in total cell lysates between JJ012 and JJ012R cells were analyzed by Western blot. (**C**) The protein levels of APLN in JJ012R/shControl, JJ012R/shAPLN#1, and JJ012R/shAPLN#2 cells were examined using Western blot. (**D**) Cells were treated with increasing concentrations of Dox (0–64 μM) for 24 h, and subsequently cytotoxicity was evaluated by an MTT assay. The percentage of cell viability is shown relative to untreated controls. * *p* < 0.05 compared with shControl group. (**E**) Cells were treated with increasing doses of Dox (0–64 μM) for 24 h and the expression levels of cleaved caspase 3 were examined using Western blot. α-Tubulin was used as a loading control. (**F**,**G**) After the miR-631 mimic was delivered into JJ012R cells, the expression levels of miR-631 and APLN were further analyzed by qPCR. * *p* < 0.05 compared with miR-NC group. (**H**) The protein expression of APLN was analyzed by Western blot. (**I**,**J**) miR-631-transfected JJ012R cells and their parental cells (JJ012 cells) were treated with 8 μM Dox for 24 h, and then the Dox susceptibility was determined by MTT and LDH assays, respectively. * *p* < 0.05 compared with miR-NC group.

## Data Availability

The data presented in this study are available upon request from the corresponding author.

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
