# Peer review of "MicroRNA-631 Resensitizes Doxorubicin-Resistant Chondrosarcoma Cells by Targeting Apelin"

_ijms, 2023, doi:10.3390/ijms24010839_

Round 1

Reviewer 1 Report

The authors in the manuscript show that doxorubicin resistant chondrosarcoma cells have increased expression of Apelin (APLN) and reduced expression of its regulator miR-631. Genetic knock-down of APLN leads to reduced cell viability of resistant cells at higher doses. Similarly, genetic knock-in of miR-631 leads to reduced APLN expression and consequent sensitization to high dose doxorubicin. The authors in general have done sound work to support their point. However, it seems like reducing APLN in the resistant cells does not sensitize them to doxorubicin like the sensitive cells, unless they are treated with a very high dose of doxorubicin. Meaning, the sensitive cells are 20% viable at 2uM doxorubicin, but the resistant cells post APLN knock-down are 20% viable only at a dose as high as 16uM. The authors should provide an explanation to this point, in order to validate their findings of APLN being important for resistant chondrosarcoma.

Additionally, it would also help if in figures 2E, 3H-I, 4D, 4I-J, authors add data for the sensitive cells as well, to answer the question - does knock down of APLN in resistant cells make them behave as the sensitive cells? This will help compare if the APLN deficient resistant cells behave the same way as the sensitive cells or not.

Author Response

  1. The authors in the manuscript show that doxorubicin resistant chondrosarcoma cells have increased expression of Apelin (APLN) and reduced expression of its regulator miR-631. Genetic knock-down of APLN leads to reduced cell viability of resistant cells at higher doses. Similarly, genetic knock-in of miR-631 leads to reduced APLN expression and consequent sensitization to high dose doxorubicin. The authors in general have done sound work to support their point. However, it seems like reducing APLN in the resistant cells does not sensitize them to doxorubicin like the sensitive cells, unless they are treated with a very high dose of doxorubicin. Meaning, the sensitive cells are 20% viable at 2uM doxorubicin, but the resistant cells post APLN knock-down are 20% viable only at a dose as high as 16uM. The authors should provide an explanation to this point, in order to validate their findings of APLN being important for resistant chondrosarcoma.

Reply:

Thanks for the reviewer's comments. Dox-resistant cells must undergo multiple mutations to adapt to Dox-containing environments, and APLN may be one of the pathways involved. Therefore, even APLN knockdown could not fully restore the same Dox sensitivity as their parental cells.

We have also added the above description to the Discussion section.

  1. Additionally, it would also help if in figures 2E, 3H-I, 4D, 4I-J, authors add data for the sensitive cells as well, to answer the question - does knock down of APLN in resistant cells make them behave as the sensitive cells? This will help compare if the APLN deficient resistant cells behave the same way as the sensitive cells or not.

Reply:

Thanks to the reviewer’s comments, we have integrated their parental cells into Figures 2E, 3H-I, 4D, and 4I-J in the revised manuscript, so that readers can better understand the Dox sensitivity between APLN knockdown-resistant cells and their parental cells.

Reviewer 2 Report

Dear Authors, 

The draft you submitted presents the results of a fascinating study to dissect the Apelin participation in Doxorubicin-resistant development in Chondrosarcoma. It was also proved that miR-631 regulates Apelin, and that overexpression of miR-631 contributes to restoring drug sensitivity. 

The project's aim is clearly presented, the methods are adequately described and the results are reported and properly discussed. 

Even considering the aim and results are clearly reported and the draft has a very high degree of revision, I would suggest a few recommendations to improve the draft before publication.  The four figures you drew up to present the whole results contain too many panels in each figure. In this reviewer's opinion, the Figures are too crowded that the readability of the same is hampered. I would suggest splitting the panels into different figures to have each of them in a proper shape and quality allowing the reader to better access your results. 

In the last, the abstract contains typing errors and references seem not to be in the right format, you should consider fixing them. 

Author Response

  1. The draft you submitted presents the results of a fascinating study to dissect the Apelin participation in Doxorubicin-resistant development in Chondrosarcoma. It was also proved that miR-631 regulates Apelin, and that overexpression of miR-631 contributes to restoring drug sensitivity. The project's aim is clearly presented, the methods are adequately described and the results are reported and properly discussed.

Reply:

We appreciate the reviewer comments.

  1. Even considering the aim and results are clearly reported and the draft has a very high degree of revision, I would suggest a few recommendations to improve the draft before publication. The four figures you drew up to present the whole results contain too many panels in each figure. In this reviewer's opinion, the Figures are too crowded that the readability of the same is hampered. I would suggest splitting the panels into different figures to have each of them in a proper shape and quality allowing the reader to better access your results.

Reply:

We appreciate the reviewer comments. we have moved Fig.1G into Fig.2 in revised manuscript to avoid the too crowded problem. For the logical order of the research, could four figures be maintained.

  1. In the last, the abstract contains typing errors and references seem not to be in the right format, you should consider fixing them.

Reply:

Thanks to the reviewer’s comments. The errors in the abstract have been corrected, and the formatting of the references has also been revised to comply with JIMS.

Round 2

Reviewer 1 Report

The authors have addressed the given comments.